# Improving the Corrosion Resistance of Aluminum Alloy by Creating a Superhydrophobic Surface Structure through a Two-Step Process of Etching Followed by Polymer Modification

**DOI:** 10.3390/polym14214509

**Published:** 2022-10-25

**Authors:** Tian Shi, Jingsong Liang, Xuewu Li, Chuanwei Zhang, Hejie Yang

**Affiliations:** 1School of Mechanical Engineering, Xi’an University of Science and Technology, Xi’an 710054, China; 2State Key Laboratory for Mechanical Behavior of Materials, School of Materials Science and Engineering, Xi’an Jiaotong University, Xi’an 710049, China

**Keywords:** aluminum alloy, superhydrophobicity, corrosion resistance, polymer modification, micro–nanostructure

## Abstract

A multifunctional aviation aluminum alloy with good superhydrophobicity and corrosion resistance was prepared by a two-step process of etching followed by polymer modification. Meanwhile, micro- and nanostructures formed on the processed sample. Compared with bare sample, the static liquid contact angle on the as-prepared sample was increased by 100.8°. Further polarization tests showed that the corrosion potential of such a sample increased, and the corrosion current density decreased obviously, thus suggesting that the corrosion resistance of the modified sample was significantly improved. The same conclusion was confirmed by subsequent impedance testing. The work is of great economic value and practical significance to enhance the corrosion resistance of aviation actuator materials and also lays a foundation for future hydrophobic application research in aeronautical engineering.

## 1. Introduction

The 7075 aluminum alloy, as an important engineering material, has excellent mechanical, physical and chemical properties [1,2]. Therefore, it is widely used in the transportation, manufacturing, construction and chemical industries [3,4]. Especially in aircraft manufacturing, it has been widely used as special actuator material [5,6]. However, the protective oxide film on aluminum alloy is easily damaged by corrosion [7]. Especially in long-term humid environments, such as overcast and rainy conditions, corrosion failure of aviation aluminum alloy will be aggravated [8]. In this way, it exerts adverse effects on the safe operation of aircraft, human life and wealth security. How to effectively improve the corrosion resistance of actuator aluminum alloy is a key scientific problem in aerospace research field.

For a long time, the application of a coating on 7075 aluminum alloy has been an effective anticorrosion method. Dhanish et al. improved the corrosion resistance of 7075 aluminum alloy through a tartaric–sulfuric acid anodized coating [9]. Yildiz et al. prepared Ni-B and Ni-W-B coatings on 7075 aluminum alloy by an electroless plating process to improve its corrosion resistance [10]. Sobolev et al. used plasma electrochemical oxidation technology to prepare a ceramic coating on 7075 aluminum alloy, which improved its corrosion resistance [11]. Wang et al. improved the corrosion resistance of 7075 aluminum alloy by preparing a micro-arc oxidation coating [12]. In addition, heat treatment, reinforcement addition, low-pressure cold spraying and laser shock peening have also been applied to improve the corrosion resistance of such aviation aluminum alloys [13,14,15]. However, some of above anticorrosion technologies are associated with a high cost. In particular, surface oxidation, electrochemical treatment, plating coating and laser operation are complex, and they may cause damage to the environment.

Hence, it is urgent and crucial to innovate new corrosion protection technology for aviation actuator alloys. Inspired by the lotus leaf effect [16], a hydrophobic surface has a better ability to repel liquid [17,18], thus reducing direct contact between the liquid and substrate. Therefore, a hydrophobic surface is expected to show good corrosion resistance. The microconvex-structure and low-surface-energy film on a lotus leaf is the direct reason for its hydrophobicity [19,20]. Mimicking this microstructure and film of the biological surface is expected to give aviation actuator aluminum alloys excellent hydrophobicity as well as corrosion resistance. Therefore, the focus of this work is to construct a microstructure and low-surface-energy polymer film on 7075 alloy. Meanwhile, from the perspective of industrial application, such a construction method should be efficient, simple, low cost and controllable.

Corrosion failure is a thorny issue that restricts Al alloy application [21]. As a research hotspot in functional realization, hydrophobic fabrication with various additive manufacturing (AM) methods offers an efficient strategy for mitigating metal corrosion [22]. However, for the AM process, the incorporation of a dissimilar metal would accelerate corrosion failure, and the additive polymer would not offer long-term structural stability [23]. Furthermore, the anticorrosion mechanisms of various wetting states have rarely been systematically investigated [24]. In this work, a two-step process of etching followed by polymer modification is used to fabricate a bionic superhydrophobic surface. Meanwhile, the preparation method is simple, and the cost is low; large-scale preparation is also realized. The efficiency of such an anticorrosion treatment for actuator aluminum alloy is greatly improved, and the operation is easy to control. The result shows that the flocculent-like microstructure was achieved on 7075 aluminum alloy. Compared with an untreated sample, the static contact angle on prepared sample increased by 100.8°. Hence, the corresponding hydrophobicity was significantly improved. Further polarization and impedance tests show that the corrosion resistance of the as-prepared sample was also markedly enhanced. Based on above treatment, this work realizes the imitation of the biological surface and its function, thus providing an effective and innovative strategy to settle the corrosion problem of aviation actuator alloys, which has great economic value and practical significance.

## 2. Materials and Methods

### 2.1. Materials

The 7075 aluminum alloy (Table 1) was purchased from Lin-Pu Material Co., Ltd., Nanyang, China. 1H,1H,2H,2H-Perfluorodecyltrichlorosilane (fluor silane) was obtained from Hong-Zhou reagent Co., Ltd., Xi’an, China. Other analytical reagents were obtained from Shi-Qiao Chemical Co., Ltd., Xuzhou, China.

### 2.2. Procedures

Firstly, the 7075 aluminum alloy was cut into a uniform block size of 10 mm × 10 mm × 2 mm by the electric spark wire cutting process. Secondly, the aluminum alloy surface was polished until smooth. After that, the polished sample was put into acetone, ethanol and ultra-pure water for ultrasonic cleaning to remove impurities, particles, organic matter and stains from the surface. Then, the sample was dried with high-purity nitrogen. Furthermore, the pretreated sample was immersed in a 3 M hydrochloric acid solution for 1, 3, 5, 7 and 9 min, respectively. After the etching process, the sample was taken out immediately and put into ultra-pure water for ultrasonic cleaning. Then, the treated sample was put into an ethanol solution of 20 mM fluor silane and soaked for 6 h at room temperature, afterward it was placed in an oven at 50 °C for 2 h. Finally, the sample was dried with high-purity nitrogen for further characterization.

### 2.3. Characterization

The surface morphology of the sample was characterized by scanning electron microscopy (SEM, QUANTA200FEG, FEI, Hillsboro, OR, USA). The surface component was analyzed using an X-ray spectrometer (EDX, Oxford Instruments, Inc. MAX, Oxford, UK). A contact angle measuring instrument (OCA20, Dataphysics Gmbh, Stuttgart, Germany) was used to test the wettability of the sample. The static contact angle is mainly measured by the sessile drop method, and a new droplet (5 μL) needs to be created first. Secondly, the baseline position should be determined. Then, the droplet profile is automatically detected. Finally, the contact angle value is calculated automatically by software. The rolling angle is measured by tilting the sample. Specifically, sample is continuously tilted until the drop rolls away from the sample. Under this condition, the sample inclination angle exactly corresponds to the resultant rolling angle. For the angle value, the average of 5 repetitions was adopted. An electrochemical workstation (CHI660D, Chenhua, Shanghai, China) was used to evaluate the corrosion resistance of the sample. During the polarization test, the scanning rate was set as 0.001 V/s, and the scanning range was tested from −1.6 to −0.6 V. In the process of electrochemical impedance spectroscopy testing, the frequency range was set from 10 mHz to 100 kHz. Before electrochemical testing, the electrode was immersed in the electrolyte for 30 min to reach a stable test status.

## 3. Results and Discussion

### 3.1. Fabrication of Hydrophobic Aviation Aluminum Alloy

Figure 1a shows the micromorphology of 7075 aluminum alloy after etching for 1 min with various magnifications. As seen, some micropits appeared on the local surface. The appearance of micropits is due to erosion dissolution of the aluminum alloy surface by hydrochloric acid. The average pit size was about 5 μm, and its distribution was dispersed and uneven. Moreover, the pit shape was irregular. In addition, compared with the rough pit structure, aluminum alloy surface without etching is relatively smooth, which is due to the initial polishing pretreatment. Figure 1b exhibits the micromorphology of aluminum alloy after etching for 3 min with different magnifications. It is seen that the etched micropit size increased obviously, and the average size was about 5–20 μm. The appearance of larger pits was caused by the expansion of the original etched pit under hydrochloric acid action with an extended etching time. It is also seen that pit distribution was relatively scattered and uneven. Meanwhile, pit shape was irregular, and many secondary micropits were formed. In addition, there were still many smooth areas that had not been etched. The reason is that the etching time of 3 min is too short for the alloy surface to be completely eroded. Figure 1c shows the micromorphology of the aluminum alloy after etching for 5 min with various magnifications. As seen, the whole surface was almost completely etched, which is rougher than the sample with shorter etching time. In this condition, the etched structures were connected together, and large-scale fabrication was realized. From the enlarged image, it is clearly seen that etched surface was uneven and hierarchical. It also exhibited microscale characteristics, and nanoscale flocculent forms at the structure’s edge. The main reason for such a structure is the existence of crystal defects in the aluminum alloy [25]. When hydrochloric acid etches the surface, the defect with higher energy dissolves preferentially under strong activation of chloride ion, while the defect with lower energy dissolves later, thus forming a flocculent structure with obvious uneven and hierarchical characteristics.

Figure 2a,b show micromorphologies with different magnifications for aluminum alloys after etching for 7 and 9 min, respectively. As seen, both surfaces were completely etched with higher roughness, and their morphologies were similar. Meanwhile, composite structures with microasperity and nanoflocculence were also clearly seen from enlarged morphologies, which were also caused by crystal defects in the aluminum alloy. In addition, through further investigation of the etched alloy, it was found that the sample thicknesses measured by the automatic optical imager equipped on the contact angle measuring instrument with etching time less than 7 min remained almost unchanged, as seen in Figure 3. However, the sample thickness after etching for 9 min was reduced by 8.6%. Therefore, it is unnecessary to investigate the etching process with a longer time, because it does not satisfy the economic index requirement for engineering materials.

In order to obtain the element composition of the etched 7075 aluminum alloy surface, the X-ray energy spectrum of the sample after etching for 7 min was measured, as shown in Figure 4. It is seen that the main elements on etched surface were aluminum, zinc, magnesium and copper. Especially, the aluminum peak was the highest. All these are consistent with the practical chemical compositions of 7075 aluminum alloy, as shown in Table 1. As seen in the table, the main contents were also aluminum, zinc, magnesium and copper (>2 wt.%). Other elements in the alloy were not detected, as seen in Figure 4, which may be due to their low contents (<0.5 wt.%). In addition, oxygen was observed owing to metal oxidation. Chlorine was also found, which is due to the chlorides obtained from chemical reactions between hydrochloric acid and metal elements in the aluminum alloy.

In order to further obtain the element distribution on the etched aluminum alloy surface, a mapping scan was conducted on the sample after etching for 7 min, and the result is shown in Figure 5a. As seen, the main element distributions in aluminum alloy were relatively uniform. The etching treatment changed the microscopic morphology on the alloy surface but had no effect on its chemical and histological characteristics. In addition, the advantage of the resultant morphology is that the etched microstructure is still a part of the matrix material, which is typical of a preparation strategy based on reducing material (Figure 5b). This is obviously different from the reported preparation strategy by adding material (Figure 5c), such as deposition [26], coating [27] and grafting [28]. For the strategy of adding material, the interface bonding between the substrate and microstructure is poor. For the reducing material strategy adopted in this work, the etched microstructure, as a part of substrate, can bear greater external loads, which is more conducive to engineering applications. Moreover, as a result of etching, an amorphous structure is most often formed, which is extremely susceptible to mechanical stress to some extent.

Figure 6a shows the wettability results of the samples after soaking in fluor silane for different times and then drying at 50 °C for 2 h. As seen, with the prolongation of polymer modification time, the static contact angle gradually increased. When polymer modification was extended to 6 h, contact angle reached the maximum value of 100.0°, indicating that the low-surface-energy polymer film had been grafted on the surface. However, on prolonging the modification time, contact angle shows a decreasing trend, which is caused by the agglomeration of fluor silane hydrolysates. Figure 6b shows wettability measurements of the samples after immersion in fluor silane for 6 h and then drying at different temperatures. As shown, when the drying temperature rises, the static contact angle gradually increases. As the drying temperature reached 50 °C, the contact angle reached the maximum value of 100.7°. Nevertheless, with the continuous increase of drying temperature, the contact angle showed a downward trend. This is because that the temperature is too high, resulting in hole damage at the defect or weak point with high energy on the polymer modification surface, which is detrimental to hydrophobicity.

The reason for the obvious increase in hydrophobicity is due to fluor silane modification. Specifically, the hydrophilic aluminum alloy surface is covered with a large number of -OH groups, while fluor silane is composed of non-polar hydrophobic long chains and polar end groups [29]. During the modification process, hydrolyzed silane undergoes a dehydration or condensation reaction with the -OH group, so that the Si-O- bond is connected to the alloy surface, and a layer of covalently bonded three-dimensional network structures is constructed [30]. Concurrently, the fluor silane film is formed, and its formation mechanism is shown in Figure 7. Fluor silane thin layer possesses high C-F bond energy, short bond distance and low polarizability, which shows strong chemical inertness and low surface energy [31]. It is precisely because of such a polymer film with lower surface energy that hydrophobicity on the modified surface is significantly increased.

To verify the rationality of the modification analysis, Fourier-transform infrared reflection (FTIR) before and after polymer modification was conducted, as shown in Figure 8. As seen, compared with the bare sample, the new characteristic peaks at 623 and 1132 cm^−1^ corresponded to Si-C and Si-O-Si bonds on the modified surface, respectively. This is due to the dehydration and condensation reactions of silane groups [32]. The new peak at 1682 cm^−1^ is related to the C-F oscillation mode. Due to the strong electron-withdrawing effect of fluorine, the silane film possesses good chemical stability, hydrophobicity and oil repellency [33]. The two characteristic peaks located at 2780 and 2842 cm^−1^ are relevant to the C-H bond, which has a repelling effect on the -OH group, thus improving the hydrophobicity of the modified film [34]. The above infrared spectrum analysis not only confirms the formation of the fluor silane film but also provides a reasonable explanation for the improvement of hydrophobicity on the modified alloy.

After obtaining the optimal polymer modification parameters, in order to test the wettability of as-prepared samples, the static water contact angles on alloy surfaces with different etching times and polymer modification were measured, as shown in Figure 9. As seen, the water contact angle on a bare sample was 59.3°, which shows obvious hydrophilic characteristics (<90°). The water contact angle increased to 102.9° for the aluminum alloy after etching for 1 min (Figure 9b), which shows hydrophobicity (>90°). However, such hydrophobicity is not obvious at this time. The same situation is found in Figure 9c. The reason for this is the biomimetic microstructure prepared on the surface after etching treatment, which enhances the hydrophobicity compared with that of the bare sample. However, fewer microstructures are prepared with short etching time, hence the hydrophobic effect is not significantly improved. When the etching time is extended to 5 and 7 min, the contact angles increased to 126.6° (Figure 9d) and 157.5° (Figure 9e), respectively. Especially, the sample etched for 7 min reached a superhydrophobic state (CA > 150° and RA < 10°). This is due to the gradually increased microstructure on the sample after the longstanding etching treatment. When the etching time was further extended to 9 min (Figure 9f), the contact angle only increased by 2.6°, and the rolling angle deceased by 0.4°, compared with the sample etched for 7 min. This is because the sample surface was completely textured after etching for 7 min. An excessively long immersion time only reduces sample thickness but has little effect on the resultant morphology. Therefore, the water contact angle on the sample surface remains relatively stable after etching for 7 min. In general, aviation aluminum alloy with excellent hydrophobic property was obtained by etching for 7 min.

### 3.2. Enhanced Corrosion Resistance of Aviation Aluminum Alloy

As mentioned above, the hydrophobicity of 7075 aluminum alloy was significantly improved after the etching process. In order to further investigate the corrosion behaviors of as-prepared samples, the polarization curves of alloys with different etching times were measured and shown in Figure 10. It is well known that the polarization curve can qualitatively characterize the corrosion ability of the sample, and also the more the corrosion potential moves to the positive direction, the stronger the corrosion resistance [35]. At the same time, the higher the corrosion current density, the worse the corrosion resistance for the sample [36]. It is clearly observed from the polarization result that with the extension of etching time, the corrosion potential gradually moves to the positive direction, which indicates that the corresponding corrosion resistance is gradually enhanced. In addition, the corrosion potentials of the samples after etching for 1 and 3 min were close, thus showing similar corrosion behaviors. Meanwhile, the corrosion potentials of the samples after etching for 7 and 9 min were close, and they also exhibited similar corrosion behaviors. These change trends of corrosion resistances for samples are consistent with the change rules of hydrophobic properties obtained in Figure 9.

In order to quantitatively characterize the corrosion behaviors of samples, the Tafel extrapolation fitting method was used to obtain the corrosion potential values of aluminum alloys with various etching times, as shown in Figure 11a. It is seen that corrosion potential for the bare sample was the lowest, thus showing the worst corrosion resistance. The corrosion potential of the sample after etching for 1 min increased to −1.353 V, suggesting an enhanced corrosion resistance. The corrosion potential of the alloy after etching for 3 min was only 0.014 V higher than that of the sample etched for 1 min, thus showing similar corrosion resistance. As etching time increased to 7 min, corrosion potential gradually increased to −1.046 V, indicating significantly enhanced corrosion resistance. When the etching time was further extended to 9 min, the corrosion potential only increased by 0.021 V, compared with the sample etched for 7 min. The variation of anticorrosion ability at each stage mentioned above is determined by the change of sample wettability. Furthermore, the corrosion current densities were also obtained after fitting, as shown in Figure 11b. As seen, the corrosion current density for bare sample was the largest, thus showing the worst corrosion resistance. The corrosion current density of the alloy with an etching time of 3 min was only 0.093 μA/cm^2^ lower than that of the sample etched for 1 min, thus showing similar corrosion resistance. With the prolongation of etching time, the corrosion current density decreased and the corresponding corrosion resistance increased gradually. By comparison, anticorrosion abilities reflected by corrosion current density and corrosion potential were consistent. In addition, the corrosion inhibition rate (*η*) was used to characterize the corrosion resistance of the sample, as follows:(1)$$\eta ={\frac{I_{\alpha }-I_{\beta }}{I_{\alpha }}}_{}$$
where *I*_α_ and *I*_β_ represent the corrosion current density of sample α and sample β. After calculation, the corrosion resistance of the sample after etching for 7 min was 83.2% higher than that of the bare sample.

Impedance spectroscopy can also be used to characterize the corrosion resistance of the sample. In order to further verify the above experimental results, impedance tests of aluminum alloys with different etching timse were conducted, and the corresponding results are shown in Figure 12. As known, in impedance spectroscopy, the larger the capacitive arc, the stronger the corrosion resistance [37,38]. It is clearly seen that the bare sample had the smallest capacitive arc size, so it possessed the worst corrosion resistance. Furthermore, with the extension of etching time, capacitive arc size gradually increased, which indicates that the corresponding corrosion resistance was also gradually enhanced. This conclusion is consistent with the polarization result in Figure 11.

In addition, a concrete equivalent circuit was used to simulate the corrosion behavior. After fitting with ZSimDemo software, the R_s_(R_f_C) circuit model (Figure 13a) can better simulate the electrochemical process of etched aluminum alloys. In this circuit, R_s_ represents solution resistance. C refers to capacitance between the etched alloy and electrolyte. R_f_ denotes the charge transfer resistance between the electrolyte and etched sample. As known, the smaller the C and the larger the R_f_, the stronger the resistance to charge transfer, and also the better the corrosion resistance [39]. The parameters of the circuit elements obtained after fitting were also obtained and are shown in Table 2. As seen, R_s_ was relatively stable in the whole process, indicating that electrochemical process remained stable in the test. At the same time, R_f_ for the bare sample was the smallest and C was the largest, which shows the worst corrosion resistance. The R_f_ and C values of the alloy etched for 3 min were similar to those of the sample etched for 1 min, thus showing similar corrosion resistance. With the prolongation of etching time, C decreased and R_f_ increased, so the corrosion resistance of etched sample gradually increased. Therefore, the conclusion obtained by impedance measurement was consistent with polarization result, which confirms the rationality of the corrosion test in this work. The bare sample possessed the worst corrosion resistance. This is because the corrosion solution is in direct contact with the metal substrate. Under the condition of solid–liquid interface (Figure 13b), chloride ion displays the strongest erosion effect. For the as-prepared hydrophobic sample in this work, the micro–nanostructure traps extensive air, which effectively promotes the formation of a solid–air–liquid interface (Figure 13c). The introduction of the air phase increases the repellence effect of the corrosive medium, and corrosion resistance is significantly improved. In addition, the air phase effectively blocks direct contact between the corrosive medium and the alloy substrate, thus weakening charge transfer, and further enhancing corrosion resistance.

## 4. Conclusions

(1)A simple, efficient, low-cost and two-step process of etching followed by polymer modification is used to prepare a bionic micro–nanostructure on 7075 aluminum alloy, which also shows excellent superhydrophobicity after polymer modification with fluor silane. Moreover, the liquid contact angle for aluminum alloy increased by 100.8° after etching process.(2)The corrosion resistance of the aluminum alloy increased by 83.2% after etching for 7 min. On one hand, hydrophobic surface weakens the charge transfer between alloy and corrosive medium. On the other hand, the hydrophobic surface significantly improves the liquid repellence ability, thus leading to significantly enhanced anticorrosion ability.(3)The work is of great economic value and practical significance to enhance the corrosion resistance of aviation actuator material and also lays a foundation for future hydrophobic application research in aeronautical engineering.

## Figures and Tables

**Figure 1 polymers-14-04509-f001:**
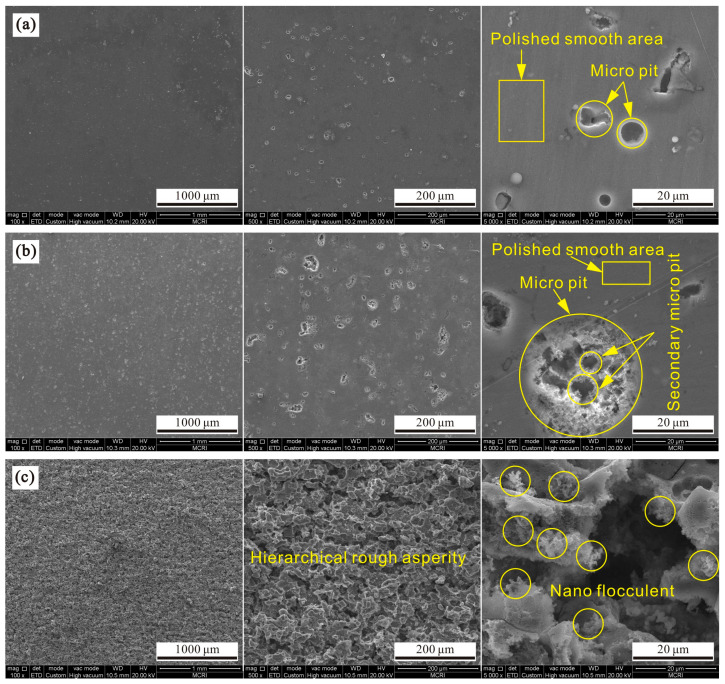
Scanning electron microscopes at various magnifications for 7075 aluminum alloys after etching for (**a**) 1 min, (**b**) 3 min and (**c**) 5 min.

**Figure 2 polymers-14-04509-f002:**
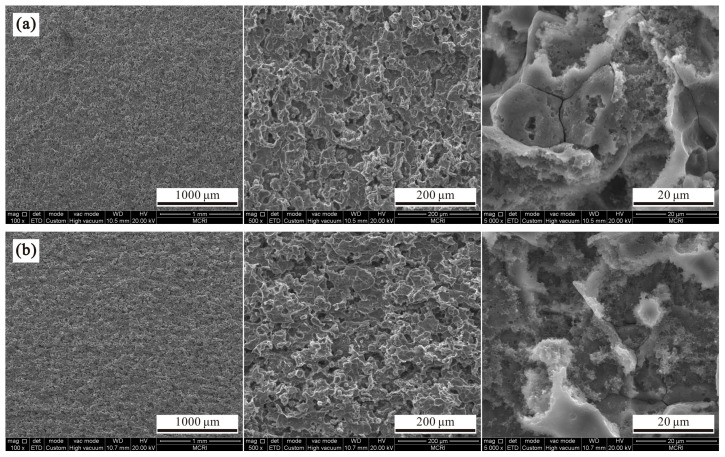
Scanning electron microscopes images at various magnifications of 7075 aluminum alloys after etching for (**a**) 7 min and (**b**) 9 min.

**Figure 3 polymers-14-04509-f003:**
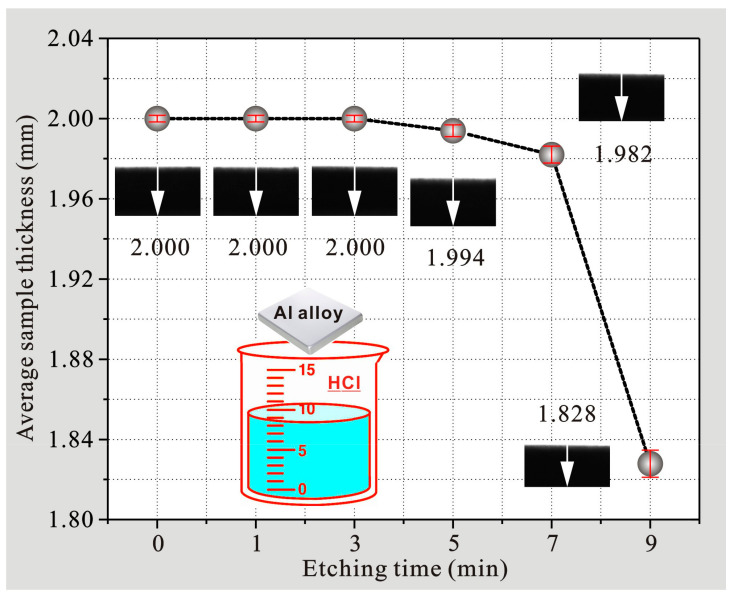
Sample thicknesses of 7075 aluminum alloys after etching for various times.

**Figure 4 polymers-14-04509-f004:**
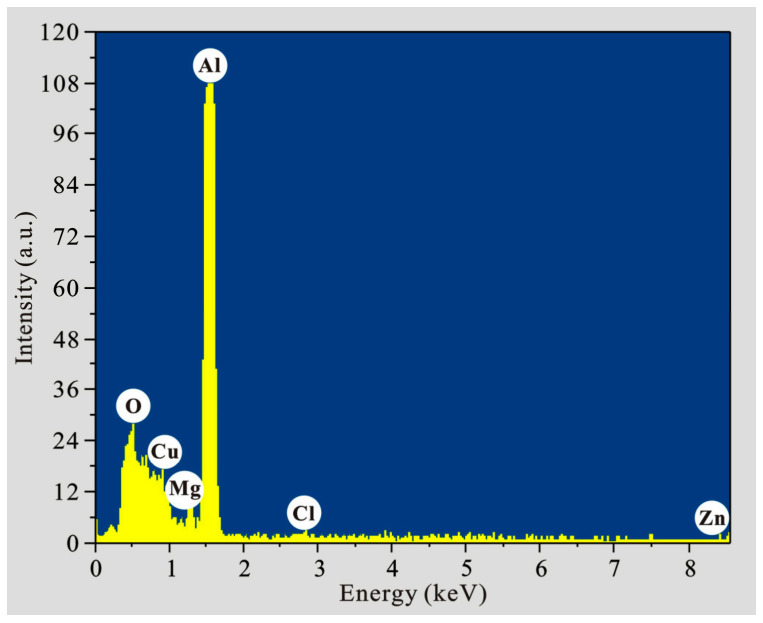
EDS analysis of 7075 aluminum alloy after etching for 7 min.

**Figure 5 polymers-14-04509-f005:**
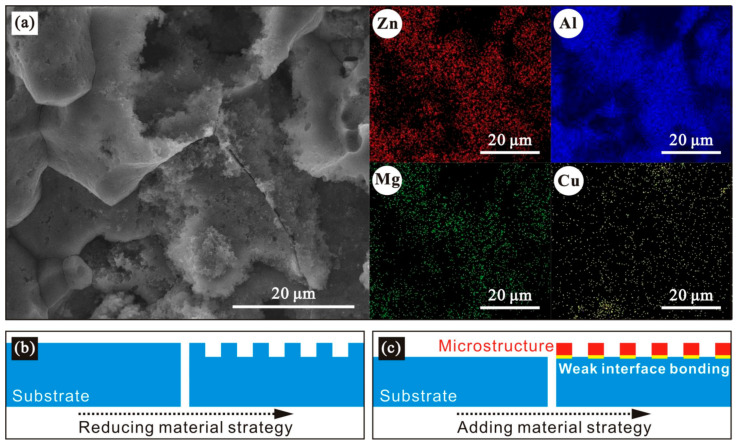
(**a**) Element distribution of 7075 aluminum alloy after etching for 7 min. Schematic diagrams of (**b**) reducing material strategy and (**c**) adding material strategy.

**Figure 6 polymers-14-04509-f006:**
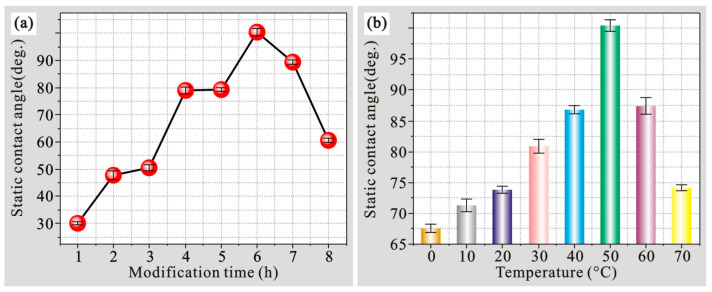
(**a**) Optimization of fluor silane modification time on aluminum alloy surface and (**b**) optimization of fluor silane drying temperature on the aluminum alloy surface.

**Figure 7 polymers-14-04509-f007:**
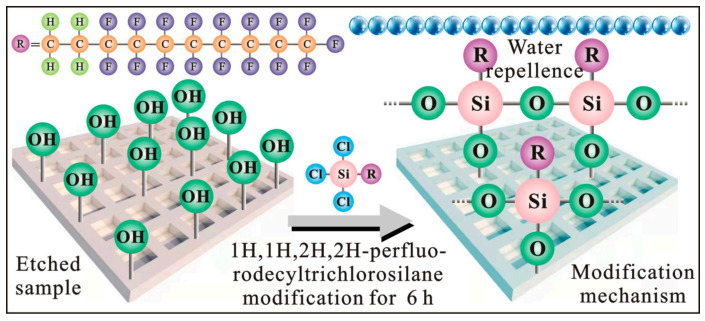
Low-surface-energy modification mechanism on the 7075 alloy surface.

**Figure 8 polymers-14-04509-f008:**
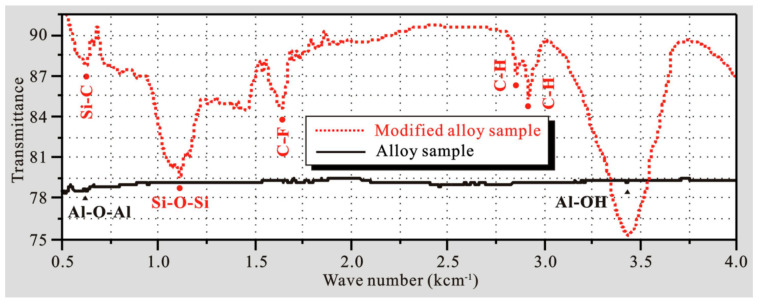
FTIR spectra on 7075 alloy surfaces before and after modification.

**Figure 9 polymers-14-04509-f009:**
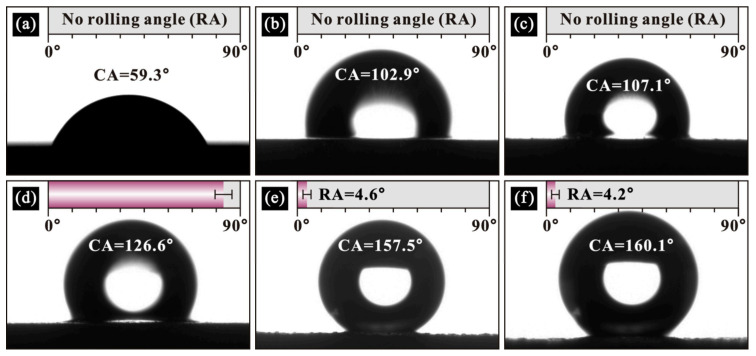
Wetting characteristics on modified 7075 aluminum alloys with different etching time: (**a**) 0 min, (**b**) 1 min, (**c**) 3 min, (**d**) 5 min, (**e**) 7 min and (**f**) 9 min.

**Figure 10 polymers-14-04509-f010:**
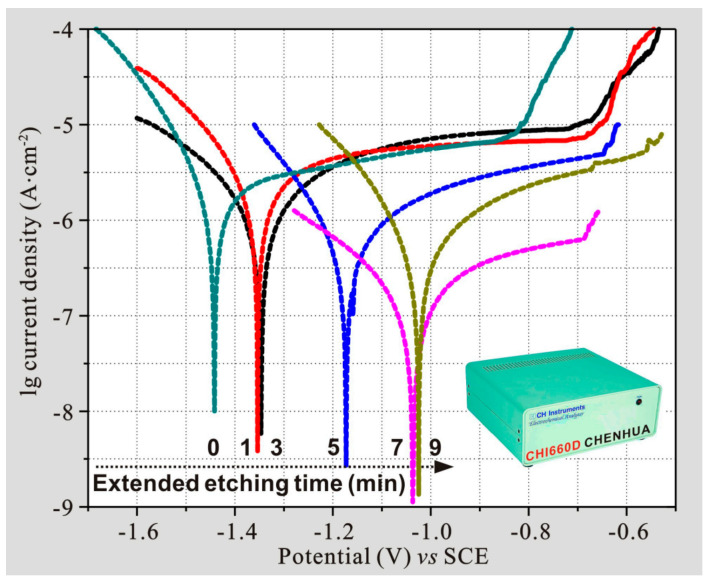
Polarization curves of modified 7075 aluminum alloys with various etching times. The insert shows the electrochemical workstation used to test the polarization curve.

**Figure 11 polymers-14-04509-f011:**
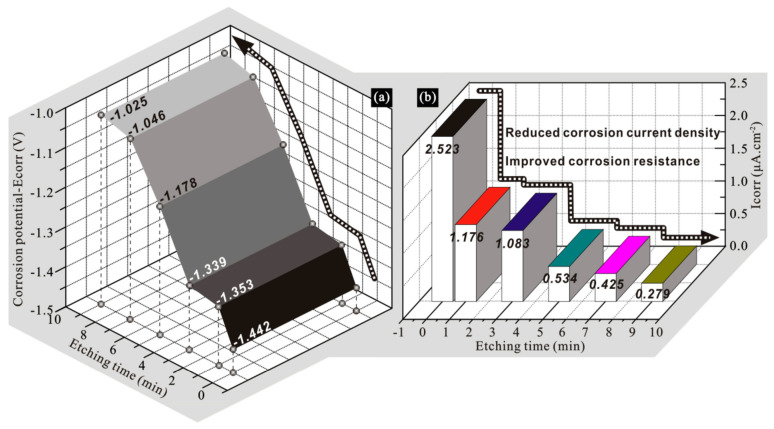
(**a**) Corrosion potentials (E_corr_) and (**b**) corrosion current densities (I_corr_) of modified 7075 aluminum alloys with various etching times.

**Figure 12 polymers-14-04509-f012:**
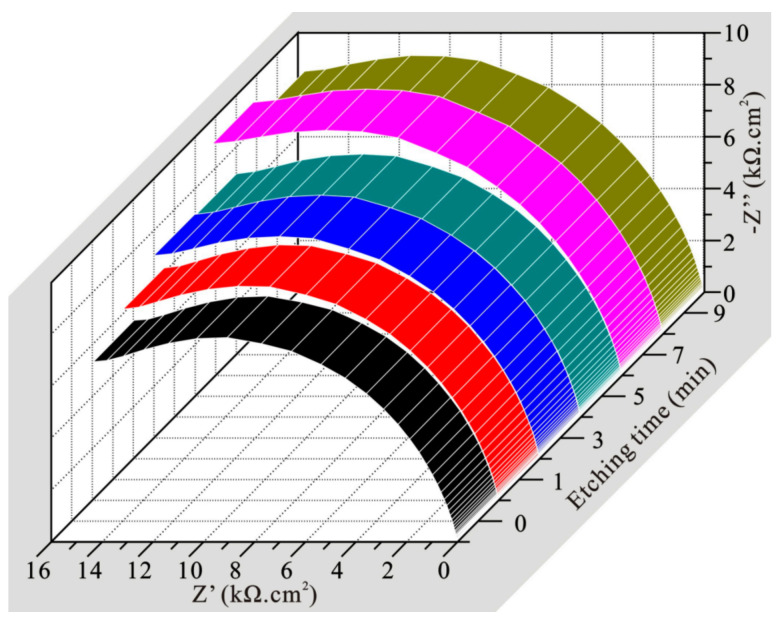
Electrochemical impedance spectra of modified 7075 aluminum alloys with various etching times.

**Figure 13 polymers-14-04509-f013:**
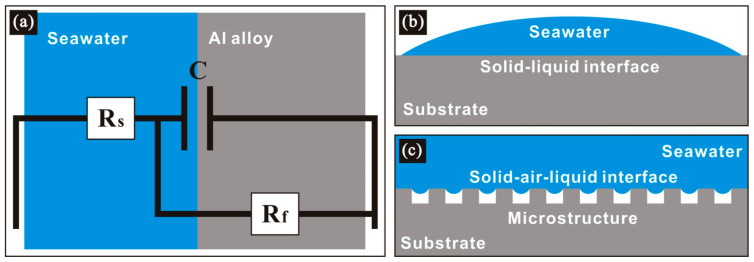
(**a**) Equivalent circuit pattern for etched 7075 aluminum alloy. Schematic diagrams of anticorrosion mechanisms for (**b**) bare aluminum alloy and (**c**) etched sample.

**Table 1 polymers-14-04509-t001:** Chemical compositions (wt.%) of 7075 aviation actuator aluminum alloy.

Element	Mg	Fe	Mn	Zn	Cu	Si	Cr	Ti	Al
wt.%	2.8	0.5	0.3	6.0	2.0	0.4	0.2	0.2	87.6

**Table 2 polymers-14-04509-t002:** Fitted electrochemical parameters of modified 7075 aluminum alloys with various etching times (min).

Samples	0	1	3	5	7	9
R_s_ (Ω.cm^2^)	7.2	6.9	6.8	6.9	7.0	7.3
R_f_ (Ω.cm^2^)	504.0	3496.6	3606.1	8374.3	10,191.8	10,274.3
C (μF.cm^−2^)	40	8.0	7.7	6.0	4.1	3.9

## Data Availability

The data that support the findings of this study are available from the corresponding author upon reasonable request.

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
