# Peer review of "Improving the Corrosion Resistance of Aluminum Alloy by Creating a Superhydrophobic Surface Structure through a Two-Step Process of Etching Followed by Polymer Modification"

_polymers, 2022, doi:10.3390/polym14214509_

Round 1

Reviewer 1 Report

Figure 1 and Figure 2 do not show the size of the micron bar.

Figure 3 How was the thickness measured?

Author Response

1. Figure 1 and Figure 2 do not show the size of the micron bar.

=>   Thanks very much for your comment. The sizes of micron bars have been re-drawn in Revised Figure 1 and Figure 2 on Page 4.

2. Figure 3 How was the thickness measured?

=>   Thanks very much for your comment. The sample thickness was measured by the automatic optical imager equipped on contact angle measuring instrument (OCA20, GERMANY DATAPHYSICS GMBH). Such description has been re-clarified and added on Page 4 in Revised Manuscript.

Reviewer 2 Report

This is an interesting paper regarding the anti-corrosion treatment of 7075 aviation aluminum alloy to create a superhydrophobic surface. The paper proposes a relatively simple and economical two-stage process of etching followed by polymer modification of surface. The paper has the scientific novelty and state-of-the-art. However, the paper needs minor revisions, and the following points should be addressed:

1. The title should be corrected for better understanding. I propose "Improving the corrosion resistance of aluminum alloy by creating a superhydrophobic surface structure through two-step process of etching followed by polymer modification". At the same time, I propose to extend this wording to the entire text of the article, including the abstract and conclusions, move the first half of the abstract to the introduction, and remove the first conclusion.

2. Table 1 should be moved from paragraph 3.1 to paragraph 2.1.

3. In references should be included the most recent and relevant reference “Qiu, C.; Liang, S.; Li, M.; Cheng, H.; Qin, W. Preparation and Application of a New Two-Component Superhydrophobic Coating on Aluminum Alloy. Metals 2022, 12, 850. https://doi.org/10.3390/met12050850".

4. Figure 7 writes about the modification of 7075 alloy surface for 12 hours, while paragraph 2.1 refers to soaking in a polymer solution for 6 hours and subsequent drying in an oven for 2 hours. Where is it correct?

5. The name of the polymer in the text of paper often changes. Why?

6. In Figure 10, log should be decimal lg.

Author Response

This is an interesting paper regarding the anti-corrosion treatment of 7075 aviation aluminum alloy to create a superhydrophobic surface. The paper proposes a relatively simple and economical two-stage process of etching followed by polymer modification of surface. The paper has the scientific novelty and state-of-the-art. However, the paper needs minor revisions, and the following points should be addressed:

1. The title should be corrected for better understanding. I propose "Improving the corrosion resistance of aluminum alloy by creating a superhydrophobic surface structure through two-step process of etching followed by polymer modification". At the same time, I propose to extend this wording to the entire text of the article, including the abstract and conclusions, move the first half of the abstract to the introduction, and remove the first conclusion.

=>   Thanks very much for your suggestion. According to reviewer comment, the title has been revised to "Improving the corrosion resistance of aluminum alloy by creating a superhydrophobic surface structure through two-step process of etching followed by polymer modification" on Page 1. Meanwhile, such wording has been extended to the entire text, including abstract and conclusions. Meanwhile, the first half of abstract has been moved to introduction on Page 1, and the first conclusion has been removed on Page 11.

2. Table 1 should be moved from paragraph 3.1 to paragraph 2.1.

=>   Thanks very much for your suggestion. According to reviewer comment, Table 1 has been moved from paragraph 3.1 to paragraph 2.1 on Page 2 in Revised Manuscript.

3. In references should be included the most recent and relevant reference “Qiu, C.; Liang, S.; Li, M.; Cheng, H.; Qin, W. Preparation and Application of a New Two-Component Superhydrophobic Coating on Aluminum Alloy. Metals 2022, 12, 850. https://doi.org/10.3390/met12050850".

=>   Thanks very much for your suggestion. According to reviewer comment, the above reference has been cited for great correlation in Revised Manuscript.

4. Figure 7 writes about the modification of 7075 alloy surface for 12 hours, while paragraph 2.1 refers to soaking in a polymer solution for 6 hours and subsequent drying in an oven for 2 hours. Where is it correct?

=>   Thanks very much for your comment. The reasonable polymer modification time is 6 hours. Such description has been re-clarified in Figure 7 on Page 7 in Revised Manuscript.

5. The name of the polymer in the text of paper often changes. Why?

=>   Thanks very much for your comment. Polymer modifier used in the text is 1H,1H,2H,2H-perfluorodecyltrichlorosilane, which is abbreviated as fluor silane. According to reviewer comment, the polymer modifier name has been uniformly revised to “fluor silane” in the whole Revised Manuscript.

6. In Figure 10, log should be decimal lg.

=>   Thanks very much for your suggestion. According to reviewer comment, log has been revised as lg in Figure 10 on Page 8.

Reviewer 3 Report

The article is devoted to the creation of a superhydrophobic aluminum surface (alloy 7075) to improve corrosion resistance. I would like to note that the main and fundamental remark is the lack of scientific novelty: the process of etching aluminum in hydrochloric acid solutions and the use of commercial substituted silanes are described in detail earlier in scientific articles (e.g. 10.1038/srep20933; 10.1016/j.apsusc.2017.04.222). Only the choice of 7075 alloy and electrochemical studies are new.

Also, the article was sent to the journal "Polymers", i.e. the main aspect should be polymer coatings and their properties. Authors need to revise the manuscript in view of the specifics of the journal.

In this edition, I cannot recommend the article for publication. The authors should revise the material and correct the following remarks:

1)    Introduction: Previously published work in this area should be reviewed.

2)    The purpose of the work does not reflect the ideas of the research and is more like conclusions.

3)    Add clause 2.1 "Materials" with a description of all reagents and materials used in the work.

4)    The authors should describe in detail the research methods.

5)    Fig3: Thickness change should be changed to mass change.

6)    Remove figure 4 and add this data to table 1. It would be interesting to trace the change in the elemental composition after different etching times and after fixing the modifier.

7)    Line 160-163. Please rephrase. As a result of etching, an amorphous structure is most often formed, which is extremely susceptible to mechanical stress.

8)    How was the contact angle measured? Fig.9. How was the rolling angle measured? Specify: the rate of inclination of the sample with respect to the horizon; drop volume; number of repetitions; scatter of values relative to the mean.

9)    The authors talk about superhydrophobicity, but the contact angle is about 100° (Fig. 6). It is necessary to provide a table with contact angle data after each modification step and various soak times in the silane solution.

10)                      Fig.7. "-Si-O-O-Si" Are you sure the mechanism is correct? Fix it. (https://doi.org/10.1021/la00056a017)

11)                      Conclusions should be rewritten.

Author Response

1)    Introduction: Previously published work in this area should be reviewed.

=>   Thanks very much for your suggestion. Corrosion failure is a thorny issue that restricts Al alloy application. As a research hotspot in functional realization, hydrophobic fabrication with various additive manufacturing (AM) methods offers efficient strategy for mitigating metal corrosion. However, for AM process, the incorporation of dissimilar metal would accelerate corrosion failure, and additive polymer could not offer a long-term structural stability. Furthermore, anti-corrosion mechanisms of various wetting states have rarely been systematically investigated. According to reviewer comment, as mentioned above, some reported works in this area as well as research aims have been concisely stated, revised, and added on Page 2.

2)    The purpose of the work does not reflect the ideas of the research and is more like conclusions.

=>  Thanks very much for your comment. The title, abstract, introduction, reference, result, discussion and conclusion have been carefully revised in order to improve systematization and scientific nature of such work. The revised work can better reflect the ideas of the research to a certain degree, and all the modified parts are marked with yellow backgrounds, as seen in the whole Revised Manuscript.

3)    Add clause 2.1 "Materials" with a description of all reagents and materials used in the work.

=>   Thanks very much for your suggestion. According to reviewer comment, 2.1 "Materials" with a description of all reagents and materials used in the work has been added on Page 2 in Revised Manuscript.

4)    The authors should describe in detail the research methods.

=>   Thanks very much for your suggestion. The research methods of structure preparation, polymer modification, morphology characterization, wettability, corrosion behavior and anti-corrosion mechanism are re-described in detail in 2.2 "Procedures", 2.3 "Characterization" and the whole Revised Manuscript.

5)    Fig 3: Thickness change should be changed to mass change.

=>   Thanks very much for your suggestion. Sample thickness can directly reflect etching degree of alloy. At the same time, etching degree can be characterized while measuring wettability. This is due to that sample thickness is measured directly by the automatic optical imager equipped on contact angle measuring instrument (OCA20, GERMANY DATAPHYSICS GMBH), which is simpler and more convenient. Such description has been re-clarified and added on Page 4 in Revised Manuscript.

6)    Remove figure 4 and add this data to table 1. It would be interesting to trace the change in the elemental composition after different etching times and after fixing the modifier.

=>   Thanks very much for your suggestion. EDS energy spectrometer is an instrument for analyzing material element. In a vacuum chamber, sample surface is bombarded with an electron beam to excite material for emitting X-ray characteristics. At the same time, according to wavelength of X-ray characteristic, the corresponding element in periodic table is obtained. EDS can provide qualitative component analysis of sample surface, as well as point, line scan and mapping analyses of specific areas. EDS is mainly used for qualitative analysis of element. It cannot accurately and quantitatively analyze chemical content of element. In this article, X-ray energy spectrum of sample after etching for 7 min is measured to obtain element composition on etched 7075 aluminum alloy, as shown in Figure 4.

7)    Line 160-163. Please rephrase. As a result of etching, an amorphous structure is most often formed, which is extremely susceptible to mechanical stress.

=>   Thanks very much for your suggestion. According to reviewer comment, the above crucial and important description has been added on Page 5-6 in Revised Manuscript.

8)   How was the contact angle measured? Fig.9. How was the rolling angle measured? Specify: the rate of inclination of the sample with respect to the horizon; drop volume; number of repetitions; scatter of values relative to the mean.

=>   Thanks very much for your suggestion. According to reviewer comment, the above crucial and important description has been added on Page 3 in Revised Manuscript.

9)    The authors talk about superhydrophobicity, but the contact angle is about 100° (Fig. 6). It is necessary to provide a table with contact angle data after each modification step and various soak times in the silane solution.

=>   Thanks very much for your suggestion. 

Figure 6a shows contact angle results of BARE SAMPLES after soaking in fluor silane for different time. As seen, with the prolongation of modification time, static contact angle gradually increases. As polymer modification is extended to 6 h, contact angle reaches the maximum value of 100.0°, indicating a HYDROPHOBIC STATE (90.0°<CA<150.0°, NOT SUPERHYDROPHOBIC ONE). Figure 6 is conducted to obtain the optimal modification parameters. In order to test wet abilities of AS-PREPARED SAMPLES (ETCH + MODIFICATION), static water contact angles on alloy surfaces with different etching time and polymer modification are measured in Figure 9. As seen, when etching time is extended to 7 min, contact angles has increased to 157.5° (Figure 9e). Especially, the sample etched for 7 min has reached a SUPERHYDROPHOBIC STATE (CA>150° and RA<10°).

10)  Fig.7. "-Si-O-O-Si" Are you sure the mechanism is correct? Fix it. (https://doi.org/10.1021/la00056a017)

=>   Thanks very much for your suggestion. According to reviewer comment, low-surface-energy modification mechanism has been revised on Page 7.

11)  Conclusions should be rewritten.

=>   Thanks very much for your suggestion. According to reviewer comment, conclusions have been rewritten on Page 11 in Revised Manuscript.

Round 2

Reviewer 3 Report

The authors did not provide an overview of the work on aluminum etching (this could improve the reader's understanding of the novelty of the work) and did not provide elemental analysis data for the line of samples: from the original to the modified with silane. However, the article has improved: the title, purpose and conclusions are consistent; the experiments are described in detail; research is consistent and logical. In general, the article can be recommended for publication.